# Cellulolytic Properties of a Potentially Lignocellulose-Degrading *Bacillus* sp. 8E1A Strain Isolated from Bulk Soil

Jakub Dobrzyński [1,*], Barbara Wróbel [1] and Ewa Beata Górska [2]

1 Institute of Technology and Life Sciences—National Research Institute, Falenty, 3 Hrabska Avenue, 05-090 Raszyn, Poland; b.wrobel@itp.edu.pl

2 Department of Biochemistry and Microbiology, Institute of Biology, Warsaw University of Life Sciences-SGGW, 02-787 Warsaw, Poland; ewa_gorska@sggw.edu.pl

* Correspondence: j.dobrzynski@itp.edu.pl

**Abstract:** Cellulolytic enzymes produced by spore-forming bacteria seem to be a potential solution to the degradation of lignocellulosic waste. In this study, several dozen bacterial spore-forming strains were isolated from soil and one of them was selected for further studies. The studied bacterial strain was identified to genus *Bacillus* (strain 8E1A) by 16S rRNA gene sequencing. *Bacillus* sp. 8E1A showed an activity of carboxymethyl cellulase (CMCase) with visualization with Congo Red-25 mm (size of clear zone). To study CMCase, filter paper hydrolase (FPase), and microcrystalline cellulose Avicel hydrolase (Avicelase) production, three different cellulose sources were used for bacterial strain cultivation: carboxymethyl cellulose (CMC), filter paper (FP), and microcrystalline cellulose Avicel. The highest CMCase ($0.617$ U mL$^{-1}$), FPase ($0.903$ U mL$^{-1}$), and Avicelase ($0.645$ U mL$^{-1}$) production of *Bacillus* sp. 8E1A was noted for using CMC (after 216 h of incubation), Avicel cellulose (after 144 h of incubation), and CMC (after 144 h of incubation), respectively. Subsequently, the cellulases' activity was measured at various temperatures and pH values. The optimal temperature for CMCase ($0.535$ U mL$^{-1}$) and Avicelase ($0.666$ U mL$^{-1}$) activity was 70 °C. However, the highest FPase ($0.868$ U mL$^{-1}$) activity was recorded at 60 °C. The highest CMCase and Avicelase activity was recorded at pH 7.0 ($0.520$ and $0.507$ U mL$^{-1}$, respectively), and the optimum activity of FPase was noted at pH 6.0 ($0.895$ U mL$^{-1}$). These results indicate that the cellulases produced by the *Bacillus* sp. 8E1A may conceivably be used for lignocellulosic waste degradation in industrial conditions.

**Keywords:** cellulases; lignocellulosic waste; spore-forming bacteria

## 1. Introduction

Every year, approximately 200 billion tons of plant biomass is produced on Earth, approximately 90% of which is lignocellulosic waste [1]. Lignocellulosic waste may be used for producing biofuels (e.g., bioethanol and biogas), gaining importance in the era of the depletion of conventional fossil fuel resources [2–4]. The main component of lignocellulosic waste is cellulose, which accounts for 40–55% of this waste dry weight [5]. Cellulose is the most common biopolymer in nature, made of β-D-glucose molecules, linked by β-1,4-glycosidic bonds. It contains crystalline and amorphous regions [6].

Complete degradation of cellulose requires an enzyme complex belonging to the class-O-glycoside hydrolases [6]. Cellulolytic enzymes include: (1) endo-β-1,4-glucanases (EC 3.2.1.4), which randomly cleave β-1,4 glycosidic bonds located in the amorphous regions of the cellulose—an example of endoglucanase is carboxymethyl cellulase, (2) exo-1,4-β-glucanases (EC 3.2.1.91), which detach glucose and cellobiose units from reducing or non-reducing ends of the cellulose chain, e.g., enzymes that degrade a microcrystalline cellulose Avicel-Avicelase, and (3) cellobiases (β-glucosidase), which convert cellobiose to glucose (EC 3.2.1.21) [7–9].



Cellulases are produced by the majority of the systematic groups of organisms: microorganisms, protists, plants, and animals, including mammals [10]. However, the industry uses mostly enzymes produced by microorganisms. The first commercial cellulases were derived from fungi, e.g., from *Trichoderma* and *Aspergillus* [11,12]. Recently, the importance of bacterial cellulases has been rising. Cellulolytic enzymes are produced by aerobic and anaerobic bacteria, e.g., bacteria of the genus *Bacillus*, *Butyrivibrio*, *Cellulomonas*, *Clostridium*, *Paenibacillus*, and *Ruminococcus* [13,14]. Of the large number of enzymes produced by resistance to harsh environmental conditions, spore-forming bacteria of the genus *Bacillus* and related genera seem to be the most interesting [15]. So far, several dozen cellulase-producing strains have been isolated and characterized, including *Alicyclobacillus acidiphilus* [16], *A. cellulosilyticus* [17], *B. sphaericus* [18], *B. subtilis* [19], *Geobacillus* sp. HTA426 [20], and *Lysinibacillus fusiformis* [21,22]. Despite that, only a small number of spore-forming bacteria can synthesize a large amount of cellulase capable of crystalline cellulose degradation in vitro [23,24].

The main factors affecting the cellulase properties of *Bacillus* and related genera, including production and activity, are temperature, time, pH, and the source of the cellulose [25,26]. This study aimed at isolating spore-forming, cellulolytic bacteria, their identification, and determination of the selected cellulases' properties, including CMCase, FPase (filter paper hydrolase), and Avicelase.

## 2. Materials and Methods

### 2.1. Cellulase-Producing Strains' Isolation

Spore-forming bacterial strains were isolated from bulk and rhizospheric soil taken from a nearly 100-year static experiment established in 1922 (located at the Experimental Station of the Faculty of Agriculture and Biology, Warsaw University of Life Sciences in Skierniewice) [27–29]. The soil for strain isolation was taken from rye and potato monoculture, rotation without legumes, and five-year rotation (including rye and potato). To isolate spore-forming bacteria (SB), 10 g of soil or fresh roots (before isolation, the roots were stripped of bulk soil by shaking) were suspended in 100 mL of saline solution (0.85% NaCl) [26,30]. Then, the suspensions were shaken for 20 min and pasteurized at 85 °C for 15 min. One milliliter of the suspension was added to Park's medium ($(NH_4)_2SO_4$—0.5 g; $KH_2PO_4$—1.0 g; KCl—0.5 g; $MgSO_4$—0.2 g; $CaCl_2$—0.1 g; 1000 mL distilled water, pH 7.4) supplemented with filter paper and incubated at 28 °C for one week. Next, 168 h-old bacterial cultures were diluted to $10^{-5}$, plated on solid Park's medium with 1% CMC, and incubated for 120 h. Single bacterial colonies were picked and then streaked on Luria–Bertani (LB) agar (BTL Ltd., Łódź, Poland). Isolates were purified by streaking repeatedly. Purified isolates were spot inoculated on Park medium with CMC and incubated for 120 h at 30 °C. Plates were flooded with an aqueous solution of 1% Congo red for 10 min at room temperature and thoroughly washed with 1 M NaCl for counterstaining the plates. Isolates with the largest clear zones were selected and screened on the liquid Park medium with filter papers to check if the strains exhibited total cellulase activity. For further analysis, isolates were preserved on LB agar slants at 5 °C for a few weeks.

### 2.2. Bacterial Strains' Identification

Initial isolate identification was carried out based on the observations of Gram- stained preparations in a light microscope and the determination of biochemical properties using bioMerieux API 50 CHB (standardized system based on 50 biochemical tests for the study of the carbohydrate metabolism of microorganisms). Bacterial isolates' identification was carried out based on 16S rRNA gene determination. The PCR template was genomic DNA-isolated using the Genomic Mini (A&A Biotechnology) from a studied bacterial culture (24 h). To amplify the 16S rRNA genes, universal primers 27F (5′-AGAGTTTGATCCTGGCTCAG-3′) and 1492R (5′-GGTTACCTTGTTACGACTT-3′) were used, as described by Kim et al. [31]. Amplification was performed in the following conditions: initial denaturation at 95 °C for 3 min, denaturation at 95 °C for 30 s (30 cycles),

annealing at 55 °C for 2 min, extension at 72 °C for 2 min, and then incubation at 72 °C for 10 min for DNA amplification. The purified PCR products were sequenced by the Sanger technique (Genomed S.A, Warsaw, Poland). The obtained sequences (forward and reverse reads) were assembled to contig (BioEdit ver. 7.2) and compared with the sequences from GenBank, EMBL (European Molecular Biology Laboratory), using BLAST (the Basic Local Alignment Search Tool). The 16S rRNA gene sequences of studied isolates have been submitted to GenBank under accession numbers: MZ482019 (15AV1), MZ481907 (15E1A), MZ481906 (24DV), MZ481905 (15AV2), MZ479750 (14AV2), MZ479749 (14AV1), and MZ479383 (8E1A).

### 2.3. Cellulase Production

Cellulase production of *Bacillus* sp. 8E1A was studied in the Park medium supplemented with three sources of cellulose: 1% CMC, Avicel cellulose (50 mg), and Whatman No. 1 filter paper (1 × 6 cm–50 mg). For this purpose, a bacterial suspension of 0.5 mL (24 h culture) was inoculated into the media (6 mL). The cultures were established in triplicate. CMCase, FPace, and Avicelase production were measured after 72, 144, 216, and 288 h of incubation (at 30 °C), using the DNS spectrophotometric method (Microplate Spectrophotometer Epoch 2, BioTek Ltd.) with calculation based on a standard curve [32]. To obtain the supernatant, the cultures were centrifuged at 10,000 rpm for 10 min at 4 °C. The reaction mixture contained 0.5 mL of supernatant, 2% CMC solution (prepared in 0.05 M potassium phosphate buffer, pH 7.0) for CMCase, 50 mg of Whatman No. 1 filter paper for FPase, 50 mg of Avicel cellulose for Avicelase, and 0.5 mL of 0.05 M potassium phosphate buffer, pH 7.0 (for FPase and Avicelase). The reference samples contained reaction mixture components plus 3 mL of DNS. The samples were incubated at 50 °C, for 30 min for CMCase, 60 min for FPase, and 90 min for Avicelase. After incubation, 3 mL of DNS was added to the solution to terminate the reaction and create a color complex. Then, all the samples were boiled at 100 °C for 5 min and cooled to room temperature. Later, the absorbance was measured at λ = 540 nm. The unit of enzyme production was defined as the amount of enzyme that could hydrolyze cellulose and release 1 µmol of glucose per one minute of reaction (U). Cellulase activity is expressed as units per mL of sample.

### 2.4. The Effect of Temperature and pH on the Cellulases' Activity

The temperature effect on the cellulases' activities of the *Bacillus* sp. 8E1A (168 h culture) was estimated using the standard procedure in the range of 30–90 °C. The effect of pH on cellulases' activities of the studied isolate (168 h culture, in Park medium) was measured by varying the pH of the reaction mixture using the following buffers (0.05 M): citrate phosphate solution, pH 3.0–6.5, sodium phosphate, pH 7.0–7.5, Tris-HCl solution, pH 8.0–9.0, and NaOH solution, pH 9.5–11.0. Further, the cellulase activity assay was performed as described for cellulase production [32].

### 2.5. Statistical Analyses

The mean values of three repetitions and the standard deviation (SD) of measurements were calculated. The results are presented graphically in the form of point and bar graphs with marked deviation posts created in the Excel software. In order to evaluate the temperature and pH effect on cellulases' activity, a one-way ANOVA was performed. The significance between means was checked with the Tukey's HSD (honest significant difference test) level of $p = 0.05$. The obtained results were processed statistically using Statistica 6.0.

## 3. Results and Discussion

### 3.1. Bacterial Strains' Identification

In this study, several dozen bacterial strains showing the cellulolytic properties were isolated, and for further assays, seven isolates that showed the highest CMCase activity

were selected. Based on the microscopic evaluation, the studied isolates were classified as Gram-positive, spore-forming bacteria.

The biochemical properties of the isolates determined by API 50 CHB are presented in Table 1. All bacterial strains were capable of decomposing D-ribose, D-galactose, D-glucose, D-fructose, D-mannitol, amygdalin, esculin, D-cellobiose, D-maltose, starch, D-melibiose, and D-sucrose. Most isolates hydrolyzed L-arabinose, D-mannose, N-acetylglucosamine, salicin, glycogen, and D-raffinose. However, none of the isolates were able to decompose erythritol, D-arabinose, L-xylose, D-adonitol, L-sorbose, dulcitol, M-α-D-mannopyranoside, inulin, D-lixose, D-tagatose, D-fucose, L-arabitol, potassium gluconate, potassium 2-ketogluconate, or potassium 5-ketogluconate (Table 1).

**Table 1.** Biochemical characteristic of isolates.

| No. | Biochemical Characteristic | Strain Number | | | | | | |
|---|---|---|---|---|---|---|---|---|
| | | 8E1A | 14AV1 | 14AV2 | 15AV1 | 15AV2 | 15E1A1 | 24DV |
| 1. | Glycerol | + | + | - | - | - | - | - |
| 2. | Erythritol | - | - | - | - | - | - | - |
| 3. | D-Arabinose | - | - | - | - | - | - | - |
| 4. | L-Arabinose | + | + | + | - | + | + | + |
| 5. | D-Ribose | + | + | + | + | + | + | + |
| 6. | D-Xylose | + | + | + | + | + | + | + |
| 7. | L-Xylose | - | - | - | - | - | - | - |
| 8. | D-Adonithol | - | - | - | - | - | - | - |
| 9. | Methyl-β-D-xylopyranoside | - | - | - | - | + | - | - |
| 10. | D-Galactose | + | + | + | + | + | + | + |
| 11. | D-Glucose | + | + | + | + | + | + | + |
| 12. | D-Fructose | + | + | + | + | + | + | + |
| 13. | D-Mannose | + | + | + | - | + | + | + |
| 14. | L-Sorbose | - | - | - | - | - | - | - |
| 15. | L-Rhamnose | - | + | - | - | - | - | - |
| 16. | Dulcitol | - | - | - | - | - | - | - |
| 17. | Inositol | + | + | - | - | + | - | + |
| 18. | D-Mannitol | + | + | + | + | + | + | + |
| 19. | D-Sorbitol | + | - | - | - | + | - | + |
| 20. | Methyl-α-D-mannopyranoside | - | - | - | - | - | - | - |
| 21. | Methyl-α-D-glucopyranoside | + | + | - | - | - | - | - |
| 22. | N-acetyl-glucosamine | - | - | + | + | + | + | + |
| 23. | Amygdalin | + | + | + | + | + | + | + |
| 24. | Arbutin | + | + | - | - | - | + | - |
| 25. | Esculin | + | + | + | + | + | + | + |
| 26. | Salicin | + | + | - | + | - | + | + |
| 27. | D-Cellobiose | + | + | + | + | + | + | + |
| 28. | D-Maltose | + | + | + | + | + | + | + |
| 29. | D-Lactose | + | + | + | + | + | + | + |
| 30. | D-Melibiose | + | + | + | + | + | + | + |

**Table 1.** *Cont.*

| No. | Biochemical Characteristic | Strain Number | | | | | | |
|-----|---------------------------|------|-------|-------|-------|-------|--------|------|
| | | 8E1A | 14AV1 | 14AV2 | 15AV1 | 15AV2 | 15E1A1 | 24DV |
| 31. | D-Sucrose | + | + | + | + | + | + | + |
| 32. | D-Trehalose | - | - | + | + | + | - | + |
| 33. | Inulin | - | - | - | - | - | - | - |
| 34. | D-Melizitose | + | + | - | - | - | - | - |
| 35. | D-Raffinose | - | + | + | - | + | + | + |
| 36. | Starch | + | + | + | + | + | + | + |
| 37. | Glycogen | + | + | + | + | + | - | + |
| 38. | Xylitol | + | + | - | - | - | - | - |
| 39. | Gentibiose | - | + | - | + | - | - | - |
| 40. | D-Turanose | + | + | - | - | - | - | - |
| 41. | D-Lyxose | - | - | - | - | - | - | - |
| 42. | D-Tagatose | - | - | - | - | - | - | - |
| 43. | D-Fucose | - | - | - | - | - | - | - |
| 44. | L-Fucose | - | - | - | - | - | - | - |
| 45. | D-Arabitol | - | - | - | - | - | - | - |
| 46. | L-Arabitol | - | - | - | - | - | - | - |
| 47. | Potassium gluconate | - | - | - | - | - | - | - |
| 48. | Potassium 2-ketogluconate | - | - | - | - | - | - | - |
| 49. | Potassium 5-ketogluconate | - | - | - | - | - | - | - |

Bacteria of the genus *Bacillus* and *Paenibacillus* studied by Akaracharanya et al. [33] also degraded sculin, D-cellobiose, D-maltose, starch, L-rhamnose, inositol, and d-sorbitol. Similar biochemical properties of SB were also obtained by Liang et al. [34].

Then, based on 16S rRNA sequence analysis, the isolates were classified to genus *Bacillus* and *Paenibacillus*: *Paenibacullus* sp. 15AV1 (MZ482019), *Bacillus* sp. 15E1A (MZ481907), *Bacillus* sp. 24DV (MZ481906), *Bacillus* sp. 15AV2 (MZ481905), *Bacillus* sp. 14AV2 (MZ479750), *Bacillus* sp. 14AV1 (MZ479749), and *Bacillus* sp. 8E1A (MZ479383).

### 3.2. Cellulolytic Properties' Detection

The highest CMCase activity measured using Congo Red was demonstrated by *Bacillus* sp. 8E1A (25 mm). Slightly lower values were observed for *Bacillus* sp. 14AV1 (18 mm) and *Bacillus* sp. 15AV2 (17 mm). The smallest clear zone was observed for *Paenibacillus* sp. 15AV2 (10 mm). The diameters of the clear zones for spore-forming bacteria reported in the literature range from 15 to 50 mm [26,35–37]. Methods using Congo Red are widespread in studies for the detection of microbial cellulase activity. However, the authors emphasized that relations between the diameter of clear zones and enzyme production were not so obvious [38]. Thus, microorganisms giving relatively low diameters of clear zone values may produce high values of CMCase and other cellulases, and vice versa. This phenomenon was also confirmed by Sadhu et al. [39] and Liang et al. [34]. Therefore, for more detailed studies, bacterial isolates' selection was based on an assay using the DNS reagent. After 168 h of incubation in Park medium with CMC, the highest CMCase and FPase production was found in *Bacillus* sp. 8E1A isolated from bulk soil of five-year rotation (data not shown).

### 3.3. Cellulase Production

Most of the studies on spore-forming cellulolytic bacteria are focused on only CMCase production [12,40]. Complete cellulose degradation requires synergistic actions of all cellulase types [26,41]. Thus, in this study, the production of the three most important cellulolytic enzymes was analyzed.

Cellulase production of *Bacillus* sp. 8E1A was maintained for 216–288 h of incubation and decreased in the late logarithmic phase of the studied bacterial strain, and similar patterns were noted by Sadhu et al. [39]. The cellulase production level was shaped by the cellulose type. The highest CMCase production occurred after 216 h of incubation in the broth with CMC (0.617 U mL$^{-1}$). Values of CMCase production measured in the broth with Avicel cellulose and FP were approximately four times lower than the production in broth with CMC after the same amount of time (Figure 1).

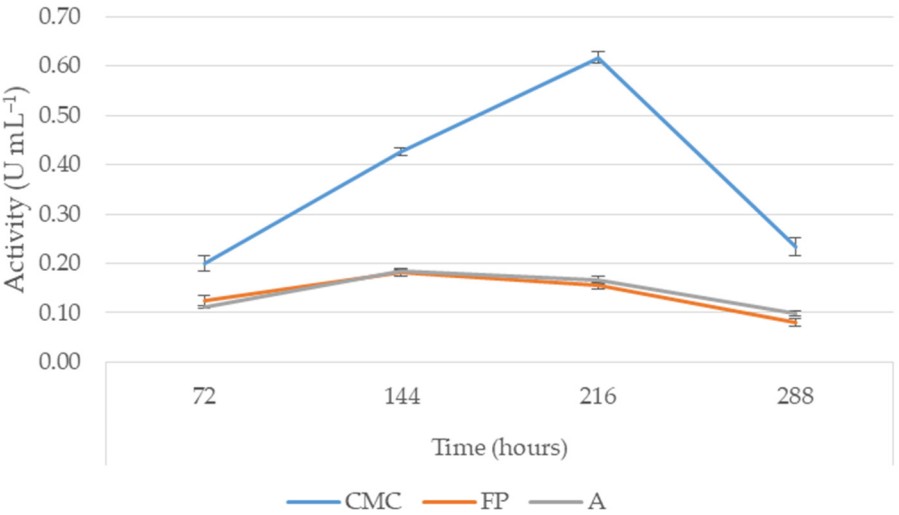

**Figure 1.** CMCase production of *Bacillus* sp. 8E1A with various sources of cellulose.

The highest Avicelase production was detected after 144 h of incubation with CMC (0.645 U mL$^{-1}$). Additionally, in bacterial cultures with other cellulose types, the highest values were found after 144 h of incubation. However, the values of bacterial cultures with FP and Avicel cellulose were nearly two times lower in comparison to bacterial cultures with CMC (Figure 2).

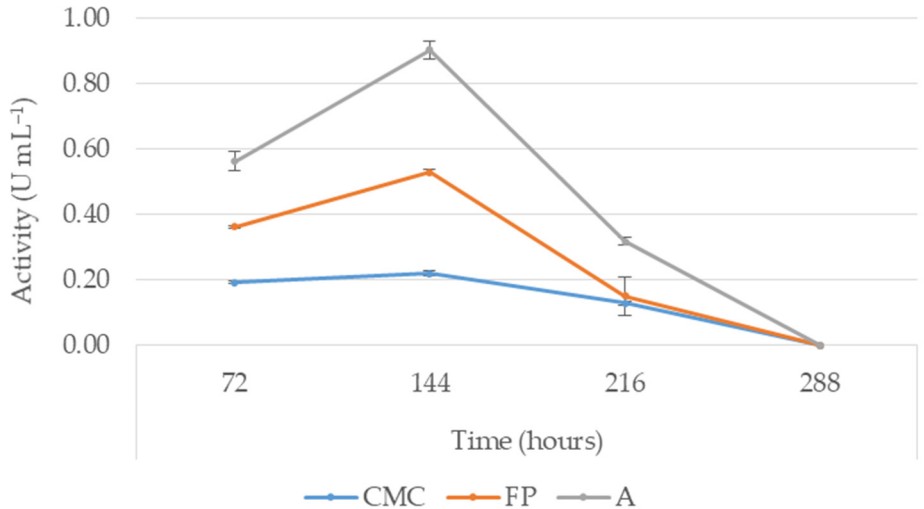

**Figure 2.** Avicelase production of *Bacillus* sp. 8E1A with various sources of cellulose.

Similar patterns were observed by other authors. Sadhu et al. [39] noted that CMC was a better indicator for CMCase and Avicelase production of *Bacillus* sp. (MTCC10046) in comparison with other carbon sources, including maltose, glucose, starch, and sucrose. Akaracharanya et al. [33] demonstrated that *Bacillus* sp. P3-1 and P4-6 had a higher cellulase production when grown in broth with CMC (value for both strains—0.015 U mL$^{-1}$) compared to broth with cellulose powder. Moreover, Thomas et al. [42] showed that CMCase production of *Bacillus* sp. SV1 was greater in broth with CMC than with Avicel and other carbon sources (e.g., chitin, glycerol, lactose, mannitol).

FPase production was detected after 72 h of incubation. The highest FPase production value was found after 144 h of incubation with Avicel cellulose—0.903 U mL$^{-1}$ (Figure 3).

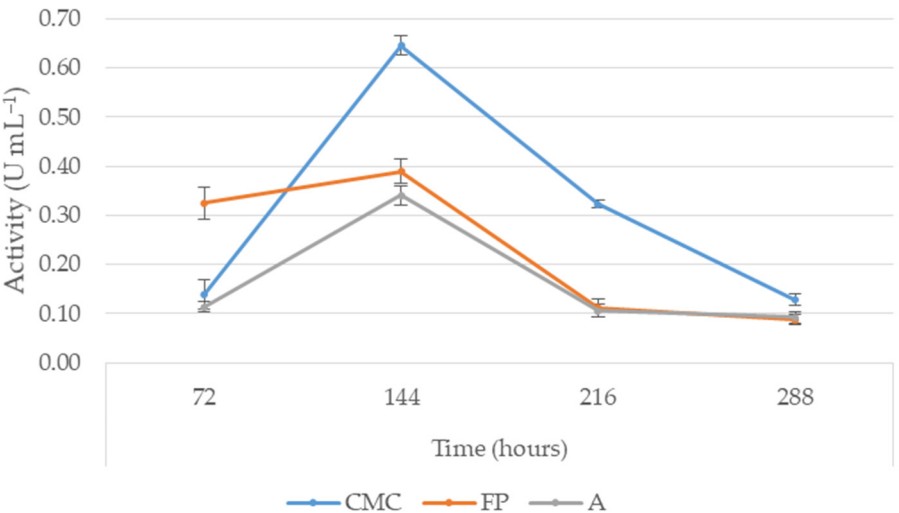

**Figure 3.** FPase production of *Bacillus* sp. 8E1A with various sources of cellulose.

Previously, Avicel cellulose has already been recognized as a substrate in the broth to increase FPase production by spore-forming bacteria. Mihajlovski et al. [26] observed a slightly higher FPase production in the Avicel broth (approximately 0.2 U mL$^{-1}$) compared to the cell-free supernatant from broth with CMC (approximately 0.1 U mL$^{-1}$) in *P. chitinolyticus* CKS1. Moreover, Kim and Kim [43] documented that thermophilic *Bacillus* sp. K-12 produced a large amount of FPase when grown in Avicel broth; however, it also produced high CMCase and Avicelase values when grown in Avicel broth. The differences between studies might be a result of culture conditions [44,45].

### 3.4. Effect of Temperature and pH on the Cellulases' Activity

Cellulases of 8E1A were active in the temperature range of 30–90 °C. There were significant differences in cellulase activity between the different temperature values ($p < 0.001$). The highest CMCase and Avicelase activity was recorded at 70 °C and was 0.535 and 0.666 U mL$^{-1}$, respectively. FPase activity was recorded at 60 °C (0.868 U mL$^{-1}$) (Figure 4).

The results of this study are similar to the results of other authors. Ladeira et al. [46] showed that the *Bacillus* sp. isolate had optimum CMCase and Avicelase activity at 70 °C. Rastogi et al. [47] showed that *Bacillus* sp. DUSELR13 had a maximum CMCase activity at 75 °C. Similar optimum temperature values for the activity of the studied enzymes were previously reported in other bacterial strains of the *Bacillus* genus [25]. Isolated from the midgut of muga silkworm, *B. pumilus* MGB05 achieved the highest activity of FPase at 50 °C [48]. Previously, Kazeem et al. [49] noted a 20 °C higher optimum temperature for FPases produced by *B. licheniformis* 2D55. Additionally, CMCase and Avicelase activity, after reaching a maximum at 70 °C, remained at a similar level up to 90 °C, which confirms their stability in high temperatures. Thermophilic cellulolytic enzymes have a large potential for application in the textile, biofuel, and agriculture industries because their manufacturing processes require high temperatures [25,50]. However, not all cellulases produced by

spore-forming bacteria have corresponding properties. In the former study, a thermophilic *Bacillus* isolate produced maximum cellulase activity at 50 °C; however, the enzyme activity significantly decreased beyond this temperature [51]. Tai et al. [52] demonstrated that CMCase activity of *Geobacillus thermoleovorans* T4 immediately decreased at temperatures above 70 °C. The studied isolate showed that the cellulase activity was in the pH range of 3 to 10. There were significant differences in the cellulases' activity between the different pH values (*p* < 0.001). The highest CMCase and Avicelase activity was recorded at pH 7 (Figure 5).

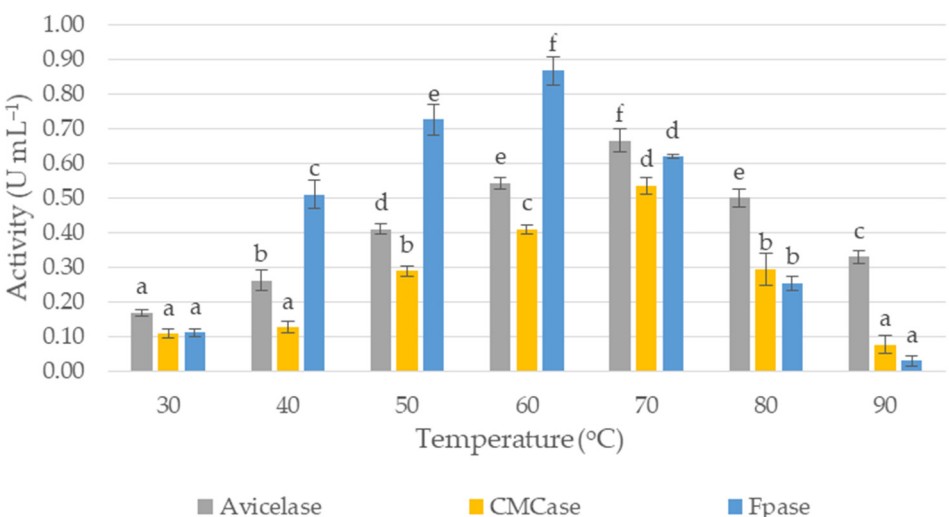

**Figure 4.** Effect of temperature on cellulase (CMCase, FPase, Avicelase) activity of *Bacillus* sp. 8E1A. Different letters above the bars (a, b, c . . . ) indicate statistically significant differences among cellulases' activity at different temperatures.

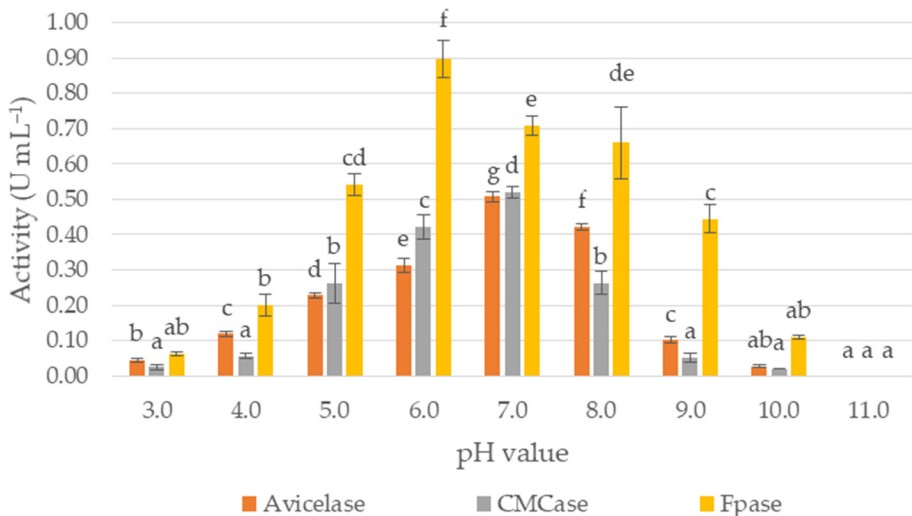

**Figure 5.** Effect of pH on cellulase (CMCase, FPase, Avicelase) activity of *Bacillus* sp. 8E1A. Different letters above the bars (a, b, c . . . ) indicate statistically significant differences among cellulases' activity at different pH values.

Similar results were reported by Ladeira et al. [46]: CMCase and Avicelase produced by *Bacillus* sp. SMIA-2 had optimum activity at pH 8.0 and 7.5, respectively. Moreover, a similar optimum pH for FPase was reported by Kazeem et al. [49], where the highest FPase activity of the *Bacillus* sp. strain was observed at pH 6.0. In the literature, there are likewise examples of cellulases showing an optimum activity at a pH below 6.0. Mihajlovski et al. [26] reported that *P. chitinolyticus* CKS1 obtained optimum Avicelase activity at pH 4.8. Similarly,

Seo et al. [25] detected the acidophilic cellulase-producing strain *B. licheniformis*. Its cellulase had stability at a pH range of 4.0–6.0. The discrepancies between studies may be caused by the large cellulase diversity of spore-forming bacteria [53].

## 4. Conclusions

In summary, the studied isolate—*Bacillus* sp. 8E1A—was able to produce all studied enzymes on each of the substrates used for the culture. The level of cellulase production depends on the cellulose type. CMCase and Avicelase production was creditably induced by the presence of CMC in the culture broth, and the highest FPase production values were recorded using Avicel cellulose as the carbon source. The wide range of optimum temperature and pH suggests that cellulases produced by *Bacillus* sp. 8E1A have the potential to be used in various industrial processes.

**Author Contributions:** Conceptualization, J.D. and E.B.G.; methodology, J.D. and E.B.G.; software, J.D. and B.W.; validation, J.D. and E.B.G.; formal analysis, J.D.; investigation, J.D.; resources, J.D.; data curation, J.D.; writing—original draft preparation, J.D.; writing—review and editing, J.D. and B.W.; visualization, B.W.; supervision, E.B.G.; project administration, E.B.G.; funding acquisition, J.D. and E.B.G. All authors have read and agreed to the published version of the manuscript.

**Funding:** This research received no external funding.

**Institutional Review Board Statement:** Not applicable.

**Informed Consent Statement:** Not applicable.

**Data Availability Statement:** Not applicable.

**Acknowledgments:** Many thanks to Katarzyna Rafalska for her help in revising the English version of the manuscript.

**Conflicts of Interest:** The authors declare no conflict of interest.

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
