# Peer review of "Cellulolytic Properties of a Potentially Lignocellulose-Degrading Bacillus sp. 8E1A Strain Isolated from Bulk Soil"

_agronomy, doi:10.3390/agronomy12030665_

Round 1

Reviewer 1 Report

Please see the file in attachment.

Author Response

Thank you for your helpful comments. We are convinced that your suggestions helped us improve our article. In line with your suggestions, we have added the missing literature references, completed the methodology, and corrected linguistic errors, and figure captions. You can find the answers to the questions below.

Is this a "first-hand" reference? The focus of this article seems different. Please check it and change if necessary.

It's hard to find "first hand" information on this issue, but we have added one reference (from the 80s), in which we found the following sentence: “A rough estimate has indicated that ca. 200 billion tons of biomass are produced in this way per year.” (without reference)

Behr, A. (1988). Carbon Dioxide as an Alternative C1 Synthetic Unit: Activation by Transition‐Metal Complexes. Angewandte Chemie International Edition in English, 27(5), 661-678.

Is 16S rRNA alone sufficient to explore "phylogeny"? What was your aim here? Just a further check of your identification? Support your answer with literature.

In our and other authors' opinion (Yarza et al. 2008), 16S rRNA is sufficient for the construction of phylogenetic trees, however, we agree that it does not contribute much to this study thus we have decided to remove this issue.

Yarza, P., Richter, M., Peplies, J., Euzeby, J., Amann, R., Schleifer, K. H., ... & Rosselló-Móra, R. (2008). The All-Species Living Tree project: a 16S rRNA-based phylogenetic tree of all sequenced type strains. Systematic and applied microbiology, 31(4), 241-250.

All other suggestions were entered into the text in the review mode.

Reviewer 2 Report

- In the abstract, they should state the design used and the cellulase activity values obtained.

- The citations should be modified to the MDPI mode.

- The introduction should contextualize the use of the variables (pH, temperature, incubation time) that affect cellulase activity by the genus Bacillus.

- In the methodology, it is important to know the characterization of the soil, the way of sample selection, and the n-value. For this, a sampling design should be proposed, which can be stratified, by clusters or simple random.

- In the isolation of microorganisms, the methodology used should be referenced.

- The methodology used to obtain cellulase activity should be referenced.

- In addition to the effect of temperature and pH, incubation time is another variable that influences cellulase activity. If the objective was to find the optimum of cellulase activity production, a central composite design with response surface analysis could be used. They should also reference where they got the values of the variables from. In addition, a more robust statistical analysis such as an ANOVA could be done. A control strain with reported cellulase activity could be used to compare yields.

The idea of the manuscript is interesting, however, the design should be improved so that the results are statistically valid and reproducible. Soil characterization and a mass balance of matter with its respective controls should be taken into account.

Author Response

Thank you for your helpful comments.  We are confident that your suggestions helped us improve our article. Below are the answers to your comments.

- In the abstract, they should state the design used and the cellulase activity values obtained.

Thank you for your suggestion, we have added more information.

- The citations should be modified to the MDPI mode.

Thank you for your comment. We have modified citations to MDPI style.

- The introduction should contextualize the use of the variables (pH, temperature, incubation time) that affect cellulase activity by the genus Bacillus.

Thank you for your comment, we have added this information.

- In the methodology, it is important to know the characterization of the soil, the way of sample selection, and the n-value. For this, a sampling design should be proposed, which can be stratified, by clusters or simple random.

Previously, these soil samples were also used to evaluate the impact of various agrotechnical treatments on the studied bacteria in soil using next-generation sequencing; the soil characterization and sampling method were included in recent study - https://www.mdpi.com/2073-4395/11/4/772. We did not place this information here because soil properties do not have any effect on cellulolytic properties of pure bacterial isolates; in this study, the main factor shaping the cellulolytic properties of bacteria was temperature, pH or incubation time during production measurements.

- In the isolation of microorganisms, the methodology used should be referenced.

Thank you for suggestion, we have added references of isolation methodology

- The methodology used to obtain cellulase activity should be referenced.

Thank you for your suggestion, we have added this reference. 

- In addition to the effect of temperature and pH, incubation time is another variable that influences cellulase activity. If the objective was to find the optimum of cellulase activity production, a central composite design with response surface analysis could be used. They should also reference where they got the values of the variables from.

One of the study aims was to check the impact of temperature and pH on the activity of studied cellulases (from Bacillus genus). In case of study about obtaining full optimization for cellulase activity, more factors should be used, e.g. different nitrogen and carbon sources, magnesium, manganese, calcium ions or different concentrations of yeast extract; and then a central composite design with response surface analysis could be used. 

Thank you for this suggestion, it is a great idea for further studies.

In addition, a more robust statistical analysis such as an ANOVA could be done. A control strain with reported cellulase activity could be used to compare yields.

Thank you for your suggestion, we have done a fisher test with homogeneous groups on the results of the temperature and pH effects on cellulolytic activity. 

According to us and other authors (Ariffin et al. 2006; Torres and Cruz 2013; Potprommanee et al. 2017), in the case of production, this type of statistical analysis is not required; the most important thing is production dynamics. 

Ariffin, H., Abdullah, N., Umi Kalsom, M. S., Shirai, Y., & Hassan, M. A. (2006). Production and characterization of cellulase by Bacillus pumilus EB3. Int. J. Eng. Technol, 3(1), 47-53.

Potprommanee, L., Wang, X. Q., Han, Y. J., Nyobe, D., Peng, Y. P., Huang, Q., ... & Chang, K. L. (2017). Characterization of a thermophilic cellulase from Geobacillus sp. HTA426, an efficient cellulase-producer on alkali pretreated of lignocellulosic biomass. PLoS one, 12(4), e0175004.

Torres, J. M. O. (2013). Production of xylanases by mangrove fungi from the Philippines and their application in enzymatic pretreatment of recycled paper pulps. World Journal of Microbiology and Biotechnology, 29(4), 645-655.

Round 2

Reviewer 2 Report

You should put the results of the anova in an annex and reference it in the results. I insist on the substrate composition since it affects the cellulase activity. (The effects of substrate composition, quantity, and diversity on microbial activity, 10.1007/s11104-010-0428-9, The effect of substrate composition on the activity of amylase and cellulase by Trichoderma harzianum strains under solid state fermentation 10.37604/jmsb.v1i2.26)

It would also be good to make a contour plot relating temperature and pH with respect to enzyme activity.

Author Response

You should put the results of the anova in an annex and reference it in the results.

  • Thank you for your comment. We have included the results of anova in a supplement (was included in the main text of manuscript, because at this moment is it no possible to adde supplement as separate file) and referred to them in the text.

I insist on the substrate composition since it affects the cellulase activity. (The effects of substrate composition, quantity, and diversity on microbial activity, 10.1007/s11104-010-0428-9, The effect of substrate composition on the activity of amylase and cellulase by Trichoderma harzianum strains under solid state fermentation 10.37604/jmsb.v1i2.26)

  • Thank you for your suggestion. The same medium was used for isolation of bacterial strains, cellulase production and for evaluation of pH and temperature effects on cellulase activity.  The medium composition is given in the section “Cellulases producing strains isolation” thus we have placed only the name of medium- Park’s medium, in the section “The effect of temperature and pH on the cellulases activity”.

It would also be good to make a contour plot relating temperature and pH with respect to enzyme activity.

  • Thank you for your suggestion, but unfortunately, we are not able to make these types of charts.
